# Trace amine-associated receptor gene polymorphism increases drug craving in individuals with methamphetamine dependence

Jennifer M. Loftis[1,2,3]*, Michael Lasarev[3,4], Xiao Shi[1,3], Jodi Lapidus[3,4], Aaron Janowsky[1,2,3,5], William F. Hoffman[1,2,3,5,6], Marilyn Huckans[1,2,3,6]

**1** Research & Development Service, VA Portland Health Care System, Portland, OR, United States of America, **2** Department of Psychiatry, Oregon Health & Science University, Portland, OR, United States of America, **3** Methamphetamine Abuse Research Center, Oregon Health & Science University, Portland, OR, United States of America, **4** Oregon Health & Science University and Portland State University School of Public Health, Portland, OR, United States of America, **5** Department of Behavioral Neuroscience, Oregon Health & Science University, Portland, OR, United States of America, **6** Mental Health and Clinical Neurosciences Division, VA Portland Health Care System, Portland, OR, United States of America

* loftisj@ohsu.edu, jennifer.loftis2@va.gov

**Data Availability Statement:** The data from this study need to be available upon request, as there are legal and ethical restrictions on sharing the data publicly. The data contain potentially sensitive

## Abstract

### Background

Methamphetamine (MA) is a potent agonist at the trace amine-associated receptor 1 (TAAR1). This study evaluated a common variant (CV) in the human *TAAR1* gene, synonymous single nucleotide polymorphism (SNP) V288V, to determine the involvement of TAAR1 in MA dependence.

### Methods

Participants (n = 106) with active MA dependence (MA-ACT), in remission from MA dependence (MA-REM), with active polysubstance dependence, in remission from polysubstance dependence, and with no history of substance dependence completed neuropsychiatric symptom questionnaires and provided blood samples. *In vitro* expression and function of CV and wild type TAAR1 receptors were also measured.

### Results

The V288V polymorphism demonstrated a 40% increase in TAAR1 protein expression in cell culture, but message sequence and protein function were unchanged, suggesting an increase in translation efficiency. Principal components analysis resolved neuropsychiatric symptoms into four components, PC1 (depression, anxiety, memory, and fatigue), PC2 (pain), PC3 (drug and alcohol craving), and PC4 (sleep disturbances). Analyses of study group and *TAAR1* genotype revealed a significant interaction for PC3 (craving response) (*p* = 0.003). The control group showed no difference in PC3 associated with *TAAR1*, while adjusted mean craving for the MA-ACT and MA-REM groups, among those with at least one

information and there is a Certificate of Confidentiality (Confidentiality Certificate No. DA-13-049) associated with this study. In order to share data, a Data Use Agreement would be needed, per guidance from the VA Portland Health Care System (VAPORHCS) and the Institutional Review Board at VAPORHCS. For submitting a data access request, please reference the MARC: Translational Service Core Biorepository (ID 03164). This repository in which the data are deposited. For more information on how to request data from the repository, please contact the corresponding author (Jennifer Loftis, email: loftisj@ohsu.edu or jennifer.loftis2@va.gov), visit the Methamphetamine Research Center (MARC) website (https://www.ohsu.edu/methamphetamine-abuse-research-center), or contact the Institutional Review Board at VAPORHCS (https://www.portland.va.gov/Research/index.asp). Mail To: VA Portland Health Care System Research and Development Service, Mail Code R&D 3710 SW U.S. Veterans Hospital Road Portland, OR 97239-2999 Phone: 503.273.5125 Fax: 503.273.5152 Email: pvamc-irb@va.gov

**Funding:** This work was supported in part by NIAAA R21AA020039 (WFH), Department of Veterans Affairs Clinical Sciences Research and Development Merit Review Program award number I0CX001558 (WFH), Department of Veterans Affairs Biomedical Laboratory Research and Development Merit Review Program award numbers 1 I01 BX002061 (JML) and 5 I01 BX002758 (AJ), DOJ 2010-DD-BX-0517 (WFH), NIDA P50 DA018165 (WFH, JML, MH, AJ), the Oregon Clinical and Translational Research Institute (OCTRI), grant number UL1TR002369 from the National Center for Advancing Translational Sciences (NCATS), a component of the National Institutes of Health (NIH), and NIH Roadmap for Medical Research. AJ is supported by the VA Research Career Scientist Program. The funders had no role in study design, data collection and analysis, decision to publish, or preparation of the manuscript. The contents of this paper do not represent the views of the U.S. Department of Veterans Affairs or the United States Government.

**Competing interests:** The authors have declared that no competing interests exist.

copy of V288V, was estimated to be, respectively, 1.55 ($p$ = 0.036) and 1.77 ($p$ = 0.071) times the adjusted mean craving for those without the *TAAR1* SNP.

## Conclusions

Neuroadaptation to chronic MA use may be altered by *TAAR1* genotype and result in increased dopamine signaling and craving in individuals with the V288V genotype.

## Introduction

Addiction to methamphetamine (MA) is a costly substance use disorder and is a growing concern, as highlighted by recent media headlines ("Meth, the Forgotten Killer, Is Back. And It's Everywhere"; [1], "Meth, Cheaper And Deadlier, Is Surging Back.", [1]). Recent epidemiological surveys have likewise documented a resurgence of MA use [2, 3] and an increase in associated fatalities [4]. MA is among the 10 most frequently mentioned drugs contributing to overdose deaths, and from 2011 through 2016, the rate of drug overdose deaths involving MA more than tripled [4]. Genetic factors contribute to risk for drug-seeking behavior, as well as to variability in addiction treatment outcomes. Polymorphisms in several genes are associated with drug dependence [5], including genes encoding opioid receptors (*e.g.*, OPRM1 [6]), phospholipase C beta 1 protein [7], and prodynorphin [8], (see also [9] for review). These gene variants may function synergistically with polymorphisms associated with comorbid neuropsychiatric symptoms that are common in addictions, such as anxiety or depression. Gene variants may also modify vulnerability to relapse and treatment response at specific stages of addiction. One leading candidate gene, which significantly influences MA use and response in animal models [10, 11], is the trace amine-associated receptor 1 (*TAAR1*) gene. Among the studied disorders in which TAAR1 plays a crucial role, drug addiction, particularly stimulant addiction, is increasingly under investigation (reviewed in [12]).

In brain, the TAAR1 appears to function as an endogenous rheostat, regulating receptor activation in neurotransmitter systems, particularly the dopaminergic system [13]. Importantly, in addition to MA's direct interactions with the dopaminergic system, MA is a potent agonist at the TAAR1 [14]. There are approximately 50 synonymous and 50 non-synonymous single nucleotide polymorphisms (SNPs) in the human *TAAR1* (dbSNP database, NCBI), and function-modifying polymorphisms of the human *TAAR1* gene are evident *in vitro* [15]. In mice, a non-functional *Taar1* allele segregates with high MA use, implying a protective role for TAAR1 function in the context of MA exposure [16]. TAAR1 has been implicated in human conditions associated with pathological monoaminergic and immune system function, including schizophrenia [17], fibromyalgia [18], migraine [19], and addictions [15, 20–23]. It is not known, however, whether differences in *TAAR1* gene sequences are associated with human MA addiction. This study evaluated a common variant (CV) in the human *TAAR1* gene, a SNP in a valine (V) codon (rs8192620 on human [GRCh38.p7] chromosome 6 at 132,645,140 bp in *hTAAR1*), occurring at amino acid position 288 and is designated V288V. The goal was to determine the involvement of *TAAR1* in humans with MA dependence—collectively investigating *TAAR1* genotype-phenotype interactions in MA addiction and recovery.

## Materials and methods

### Research participants

Participants were recruited from Portland, Oregon (OR) area addiction treatment centers and the community through word of mouth and *via* study advertisements posted in clinics,

websites, and newspapers. Individuals were enrolled into one of five groups: 1) control (CTL) group (n = 31): adults with no lifetime history of dependence on any substance other than nicotine or caffeine; 2) MA-active (MA-ACT) group (n = 13): adults actively using MA and currently meeting criteria for MA dependence; 3) MA-remission (MA-REM) group (n = 19): adults in early remission from MA dependence $\geq$ 1 month and $\leq$ 6 months; 4) active polysubstance dependence (POLY-ACT) group (n = 11): adults actively using and dependent on MA and at least one other substance (other than caffeine or nicotine); and 5) polysubstance remission (POLY-REM) group (n = 32): adults in early remission (abstinence $\geq$ 1 month and $\leq$ 6 months) from dependence on MA and at least one other substance (other than caffeine or nicotine). (*Note*: *Although the currently accepted terminologies are MA use disorder and polysubstance use disorder, as per the DSM-5, the diagnostic categories were previously termed MA dependence and polysubstance dependence, and these terms are used when referring to the research participants described in this paper*). This study was reviewed and approved by the Veterans Affairs Portland Health Care System and Oregon Health & Science University Institutional Review Boards. The procedures followed were in accordance with the ethical standards of these institutional review boards and with the Helsinki Declaration, as revised in 2004. Written informed consent was received from participants prior to inclusion in the study.

General exclusion criteria included: 1) history of major medical illness (by subject self-report during the structured interview) or current use of medications that are likely to be associated with serious neurological or immune dysfunction [e.g., stroke, traumatic brain injury, human immunodeficiency virus (HIV) infection, primary psychotic disorder or active psychosis, immunosuppressants, antivirals, benzodiazepines, opiates, stimulants, antipsychotics, anticholinergics, antiparkinson agents], 2) visible intoxication or impaired capacity to understand study risks and benefits or otherwise provide informed consent, and 3) based on the Diagnostic and Statistical Manual of Mental Disorders-Fourth Edition (DSM-IV) [24] with confirmation by the Mini International Neuropsychiatric Interview questionnaire (MINI) [25], meets criteria for past or current manic episode, schizophrenia, schizoaffective disorder, or other psychotic disorder. (Note that a history of temporary substance-induced psychosis was acceptable for participants in the substance use groups, as long as they did not currently meet criteria for a psychotic disorder, and they did not meet criteria for a psychotic disorder prior to active substance abuse).

Additional exclusion criteria for the control group included: 1) meets criteria for lifetime history of dependence on any substance (other than nicotine or caffeine dependence) based on the DSM-IV [24] with confirmation by the MINI [25], 2) heavy alcohol use as defined by the National Institute on Alcohol Abuse and Alcoholism (women: average alcohol use $\geq$ 7 standard drinks weekly for $\geq$ 1 year; men: average alcohol use $\geq$ 14 standard drinks weekly for $\geq$ 1 year [26]), 3) use of marijuana > 2 times per month, 4) regular use of other addictive substances, 5) on the day of the study visits, tests positive on a urine drug analysis for any addictive drugs, including alcohol and marijuana, and 6) education beyond an associate's degree.

Inclusion criteria for the MA dependent groups included: 1) DSM-IV criteria [24] (with confirmation by the MINI [25]) for MA dependence, 2) MA use > 2 days per week for > 1 year, and 3) no dependence [DSM-IV criteria [24] with confirmation by the MINI [25]) on other substances. Inclusion criteria for the polysubstance-dependent groups included: 1) DSM-IV criteria (with confirmation by the MINI [25]) for dependence on MA and at least one other substance (other than caffeine or nicotine), and 2) MA use > 2 days per week for > 1 year. Inclusion criteria for the active groups included: 1) meets DSM-IV criteria [24] for dependence on MA with confirmation by the MINI [25], 2) average MA use of $\geq$ 2 days per week for $\geq$ 1 year, and 3) last use of MA was $\leq$ 2 weeks ago. Inclusion criteria for the

remission groups included: 1) all criteria for the active groups, except last use of MA and other substances (other than caffeine and nicotine) was $\geq 1$ and $\leq 6$ months ago, and 2) on the day of the study visit, does not test positive for any addictive drugs, including alcohol and marijuana.

## Procedures

Participants completed a structured clinical interview (MINI), urine drug screen [Uscreen 6 Panel Drug Test Cups (RapidDetectINC, Poteau, OK, USA)], HCV and HIV antibody screen, neuropsychological tests and provided a blood sample. Neuropsychological measures included the Patient Health Questionnaire (PHQ-9) to assess depressive symptoms [27], Generalized Anxiety Disorder 7-item scale (GAD-7) [28], Prospective and Retrospective Memory Questionnaire (PRMQ), a 16-item measure which records the frequency with which subjects report problems with aspects of everyday memory functioning) [29], visual analog scales (VAS), 10 cm lines on which subjects mark alcohol and other drug cravings on a 0 to 100 scale (*e.g.*, [30]), Fatigue Severity Scale (FSS), 9-item scale to rate self-reported tiredness [31], Brief Pain Inventory (BPI) [32], and Pittsburgh Sleep Quality Index (PSQI), a 10-item scale to investigate several aspects of sleep quality [33]. Interviews were conducted in clinical space at the VA Portland Health Care System (VAPORHCS) by three trained research associates under the direction of neuropsychologist Marilyn Huckans, PhD. Diagnoses and interpretation of responses to structured interviews were discussed at weekly meetings with Dr. Huckans, but no formal inter-rater reliability measures were calculated. Blood was drawn from non-fasting participants at rest by one-time venipuncture in clinic at the VAPORHCS. Samples were collected in cell preparation tubes (BD Vacutainer Systems, Franklin Lakes, NJ, USA) containing 1 ml of 0.1M sodium citrate solution. Peripheral blood mononuclear cells (PBMCs) were isolated per standard operating procedures. PBMCs were washed twice in RPMI/5% FBS and then counted before freezing and storage in liquid nitrogen.

## DNA extraction, PCR amplification, and TAAR1 sequencing

DNA was extracted using Puregene kits (Qiagen Inc., Germantown, MD, USA). PCR was performed on an Applied Biosystems (ABI) 9600 thermocycler using *TAAR1* specific primers (Forward 5' CCTGATTATGGATTTGGGAAAA 3' Reverse 5' TCATAAAGGTCAGTACCCC AGA 3') using Amplitaq gold 360 (Applied Biosystems, Foster City, CA, USA). DNA sequencing was performed using ABI BigDye v3.1 cycle sequencing reagents and analyzed on an ABI 3130XL Genetic Analyzer. DNA extraction quality analysis of 12 samples found that the 260/280 ratios were between 1.7 and 2.0 and all 260/230 ratios were greater than 1.5; the yields ranged from 7–27 μg at a concentration between 37 and 127 ng/μl. The variation in yield was due to differences in the initial cell numbers. Average yield was 19 μg and 92 ng/μl.

## TAAR1 polymorphism analysis

The *TAAR1* variants were detected by direct Sanger sequencing and analyzed by using Sequencher 5.0 software (Gene Codes Corporation, Ann Arbor, MI, USA). Briefly, the sequencing data were trimmed to remove the low quality area. The variants were identified by comparative sequence alignments among the samples to the human *TAAR1* reference sequence (NCBI reference sequence NC_000006.12: c132646026-132644898/NM_138327.2). The potential heterozygotes were analyzed by the ratio of primary and secondary peaks and manually validated. The candidate variants from the sample sequence were determined by comparison with the established NCBI dbSNP database and Ensembl SNP database.

### *In vitro* expression and function of CV and wild type (WT) TAAR1 receptors

Chinese Hamster Ovary (CHO-K1) cells in culture were transfected with 2.5 μg/10 cm dish of either WT or CV venus-tagged cDNA. Protein expression and fluorescence were measured by a modification of our previously described methods [15]. EC50 values for the effects of β-phenethylamine (β-PEA) on cAMP production were also determined according to Shi et al. (2016). Secondary WT and CV mRNA structure was determined using RNAstructure tool software (http://rna.urmc.rochester.edu/RNAstructureWeb/).

## Statistical analysis

For *in vitro* experiments, differences in light intensity between WT and CV *TAAR1*-transfected cells were analyzed by Student's t-test. Dose-response curves for cAMP accumulation were analyzed by nonlinear regression using GraphPad Prism (San Diego, CA, USA). Between-group differences were assessed by one-way analysis of variance (ANOVA) or two-way ANOVAs of genotype by β-PEA concentration, followed with Tukey's multiple comparison test using GraphPad Prism software (San Diego, CA, USA). *P*-values < 0.05 were considered significant. To evaluate the relationship between *TAAR1* genotype and neuropsychiatric function, principal component analysis (PCA) was applied to the correlation matrix of pre-specified neuropsychiatric characteristics of interest. Components were retained until at least 80% of the overall variance was accumulated. Those retained were rotated (orthogonal Varimax) to clarify the structure and limit neuropsychiatric characteristics from loading on multiple components simultaneously. PC scores were computed and correlated against individual neuropsychiatric characteristics to facilitate interpretation and identify dominant characteristics within each PC. Scores were then used as the response of interest in separate generalized linear models (GLM) with study group, *TAAR1* genotype, and the interaction between these factors as predictors of primary interest, while adjusting for key demographic variables found also to be associated with the response. Exploratory plots revealed skewness in PC scores and increasing standard deviation proportional to the mean, so the GLM utilized a gamma distribution (with log link) after adding a small amount to all values sufficient to ensure all values were positive (shifted +1.4). Robust standard errors were used for estimation and testing to guard against potential mis-specification of the underlying distribution.

## Results

### *In vitro* expression and function of CV and WT TAAR1 receptors

There was greater TAAR1 protein expression over time in the cells transfected with cDNA for the CV compared to WT receptor (**Fig 1A, 1B and 1C**). Note that at all post transfection time points, the CV receptor was expressed at higher levels than the WT receptor, indicating that this was not a transient kinetic phenomenon. Protein values and cell number were the same across the constructs. Production of cAMP was higher in the CV- compared to WT-transfected cells in response to β-PEA, but EC50 values did not differ (230 nM vs 237 nM for WT and CV, respectively, **Fig 1D and 1E**). However, there was a significant increase in the maximal cAMP response to agonist stimulation in the cells expressing the CV, compared to the cells expressing the WT receptor (**Fig 1E**). Determination of mRNA structure revealed that the A → T substitution in the valine codon occurred near the base of a hairpin loop, but the predicted secondary structures were identical.

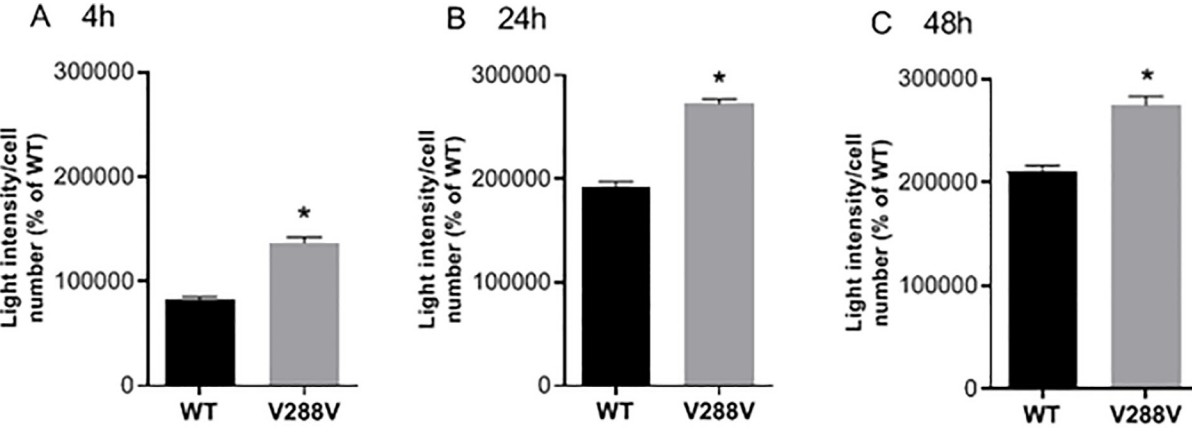

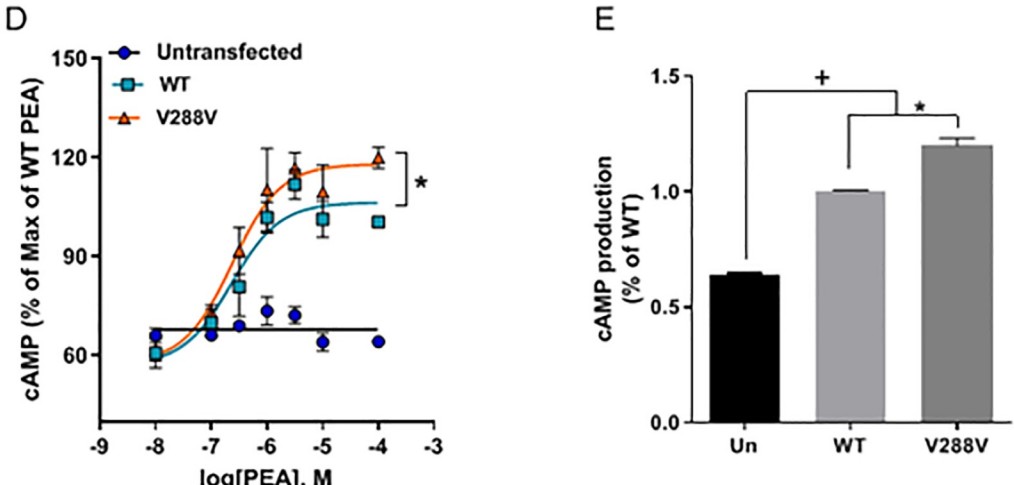

**Fig 1. The common variant, V288V has higher expression and function in transfected cells.** Cells were transfected and processed as described in the text, and fluorescence was measured at 4 hours (**A**), 24 hours (**B**), and 48 hours (**C**). Expression of V288V was higher at all time points compared to expression of the reference sequence (WT) (t-test, * $p < 0.005$). **D**) β-phenthylamine (β-PEA)-stimulated cAMP production indicated a higher maximal response but no change in EC50 values across genotypes. Two-way ANOVA detected a significant genotype by β-PEA concentration interaction [$F(12,41) = 5.882$; $p < 0.0001$), * $p < 0.005$, comparing WT and V288V variant]. **E**) One-way ANOVA, followed with Tukey's multiple comparison test compared cAMP levels between groups (*$p < 0.005$, comparing WT and V288V variant; + $p < 0.005$, comparing untransfected (Un) cells and cells expressing the TAAR1 variants.

## Demographic and clinical characteristics

Demographic and clinical characteristics of study participants are summarized in **Table 1**. The reported allelic frequency of the *TAAR1* synonymous V288V SNP in the general population is 22% [34] and was 29% (95% CI: 23% - 36%) in our sample. The distribution of V288V genotypes did not statistically differ among study groups (**Table 1**).

## *TAAR1* V288V genotype and neuropsychiatric symptoms

Common risk factors can contribute to both substance use disorders and the adverse effects on mental health. We sought to determine whether the *TAAR1* V288V genotype was associated

**Table 1. Demographic and clinical characteristics of study participants with and without a history of dependence on methamphetamine and other substances.**

| | *TAAR1* V288V genotype | | | | |
| --- | --- | --- | --- | --- | --- |
| **Characteristic** | **Negative (n = 54)** | | **Positive (n = 52)** | | ***p*-value** |
| Study group (n, %) | | | | | 0.431 |
| CTL | 17 | 31.5 | 14 | 26.9 | |
| MA-ACT | 8 | 14.8 | 5 | 9.6 | |
| MA-REM | 11 | 20.4 | 8 | 15.4 | |
| POLY-ACT | 3 | 5.6 | 8 | 15.4 | |
| POLY-REM | 15 | 27.8 | 17 | 32.7 | |
| Age [yrs] (mean, sd) | 37.6 | 10.4 | 38.6 | 12.3 | 0.644 |
| Sex (n, %) | | | | 0.485 | |
| Male | 35 | 64.8 | 37 | 71.2 | |
| Female | 19 | 35.2 | 15 | 28.8 | |
| Ethnicity (n, %) | | | | | 0.857 |
| Other | 9 | 16.7 | 8 | 15.4 | |
| White | 45 | 83.3 | 44 | 84.6 | |
| Yrs of education (mean, sd) | 12.5 | 1.5 | 12.4 | 1.5 | 0.374 |
| BMI [kg/m^2] (mean, sd) | 27.7 | 4.6 | 25.9 | 4.6 | 0.056 |
| Prescription meds. (n, %) | | | | | 0.629 |
| No | 35 | 64.8 | 36 | 69.2 | |
| Yes | 19 | 35.2 | 16 | 30.8 | |
| Current medical cond. (n, %) | | | | | 0.880 |
| No | 20 | 37.0 | 20 | 38.5 | |
| Yes | 34 | 63.0 | 32 | 61.5 | |
| Current smoker (n, %) | | | | | 0.395 |
| No | 14 | 26.9 | 18 | 34.6 | |
| Yes | 38 | 73.1 | 34 | 65.4 | |

Abbreviations: ACT, active; BMI, body mass index; CLT, control; MA, methamphetamine; POLY, polysubstance

with addiction-related symptoms, specifically, neuropsychiatric characteristics involving depression, anxiety, pain, sleep, drug and alcohol craving, and cognitive function. Principal component analysis resolved the 10 characteristics into four principal components (PC) that combine characteristics into distinct patterns: PC1 (mood/cognition) was a combination of depression, anxiety, memory, and fatigue; PC2 was dominated by pain; PC3 involved craving; and PC4 was driven by sleep disturbances (**Table 2**).

Scores for each principal component were analyzed to determine whether differences associated with *TAAR1* genotype varied by study group (*i.e.*, *TAAR1* x group interaction). For PC3 (craving response), and controlling for age, years of education, ethnicity, and sex, there was a significant [$X^2$(4df) = 15.82, $p$ = 0.003] interaction between *TAAR1* and study group. Control participants showed no difference in PC3 associated with *TAAR1* (fold change = 0.99, $p$ = 0.94), while the adjusted mean craving response for individuals in the MA-ACT and MA-REM groups, among those with at least one copy of the *TAAR1* V288V SNP (*TAAR1* positive group), was estimated to be, respectively, 1.55 (95% CI: 1.03–2.35; $p$ = 0.036) and 1.77 (95% CI: 0.95–3.27; $p$ = 0.071) times the adjusted mean response for those without the *TAAR1* SNP (*TAAR1* negative group). Participants in the POLY-REM group evidenced a similar, but non-significant trend; the adjusted mean craving response for the *TAAR1* positive group was estimated to be 1.34 (95% CI: 0.90–2.01; $p$ = 0.152) times the adjusted mean response for the *TAAR1* negative group. In the POLY-ACT group, the direction reversed, such that the

**Table 2. Pattern structure (loadings) of addiction-related symptoms identified through PCA[a].**

| Symptom (measure) | PC1 | PC2 | PC3 | PC4 |
|---|---|---|---|---|
| Depression (PHQ-9) | 0.491 | -0.058 | 0.048 | -0.001 |
| Anxiety (GAD-7) | 0.470 | -0.090 | 0.078 | 0.109 |
| Memory (PRMQ) | 0.425 | 0.025 | 0.018 | -0.080 |
| Fatigue (FSS) | 0.402 | 0.127 | -0.107 | -0.032 |
| Pain severity (BPI) | 0.023 | 0.706 | -0.029 | -0.102 |
| Pain interference (BPI) | -0.030 | 0.685 | 0.029 | 0.110 |
| Alcohol craving (VAS) | -0.256 | 0.004 | 0.834 | -0.031 |
| MA craving (VAS) | 0.229 | -0.013 | 0.365 | 0.084 |
| Other craving (VAS) | 0.255 | 0.050 | 0.350 | -0.460 |
| Sleep disturbance (PSQI) | 0.107 | 0.043 | 0.163 | 0.859 |
| Variance | 3.55 | 1.85 | 1.78 | 0.86 |
| Var. (%) | 36 | 18 | 18 | 9 |
| Cum. % | 36 | 54 | 72 | 80 |

[a]The dominant neuropsychiatric symptoms are highlighted with gray background. Abbreviations: PCA, principal component analysis; PHQ-9, Patient Health Questionnaire; GAD-7, Generalized Anxiety Disorder 7-item scale; PRMQ, Prospective and Retrospective Memory Questionnaire; FSS, Fatigue Severity Scale; BPI, Brief Pain Inventory; VAS, visual analogue scale; PSQI, Pittsburgh Sleep Quality Inventory

adjusted mean response for the *TAAR1* positive group was lower (fold change = 0.61) than the mean response for *TAAR1* negative group (95% CI: 0.41–0.91; $p = 0.016$); however, the POLY-ACT group was limited to only three WT participants and two of these were outliers (Studentized residuals > 4.5). Therefore, a secondary analysis was performed, omitting the outliers and with the POLY-ACT and POLY-REM groups combined. In addition, as the skewed distribution of the craving component resulted primarily from inclusion of the craving values of the CTL group, the CTL groups were omitted from the model. This reanalysis, using a general linear model with genotype and group as main effects and age, sex, ethnicity, and education as covariates, found no interaction between *TAAR1* and diagnostic group. When reanalyzed omitting the interaction term, there were significant main effects of *TAAR1* ($p < 0.025$) and group ($p < 0.002$). Thus, inclusion or exclusion of the outliers did not affect the overall conclusions. The main effect of *TAAR1* genotype had a Cohen's d = 0.58, indicating a moderate effect size in the multiple regression analysis (see S1 Statistical Supplement).

Further testing revealed that the *TAAR1* genotype effects in the MA-ACT and MA-REM groups (*i.e.*, 1.55- and 1.77-fold increases) did not differ significantly [$X^2$ (1df) = 0.11, $p = 0.75$], implying a shared underlying effect on craving for those with MA dependence, either active or in early remission. Having at least one copy of the *TAAR1* V288V SNP was associated with a 1.68 (95% CI: 1.14–2.47, $p = 0.009$) fold increase in the adjusted mean craving response as characterized by PC3. This suggests participants with *TAAR1* V288V SNP in the MA-ACT and MA-REM groups reported higher aggregated levels of craving (for alcohol, MA, and other substances) than participants without the *TAAR1* V288V SNP (**Fig 2**).

## Discussion

This is the first study to describe the effects of an expression-modifying polymorphism of the *TAAR1* gene on characteristics of humans with a MA use disorder. These important findings provide evidential support for the role of *TAAR1* genotype in the craving response. The CV SNP resulted in increased functional TAAR1 protein production in cell culture although it is

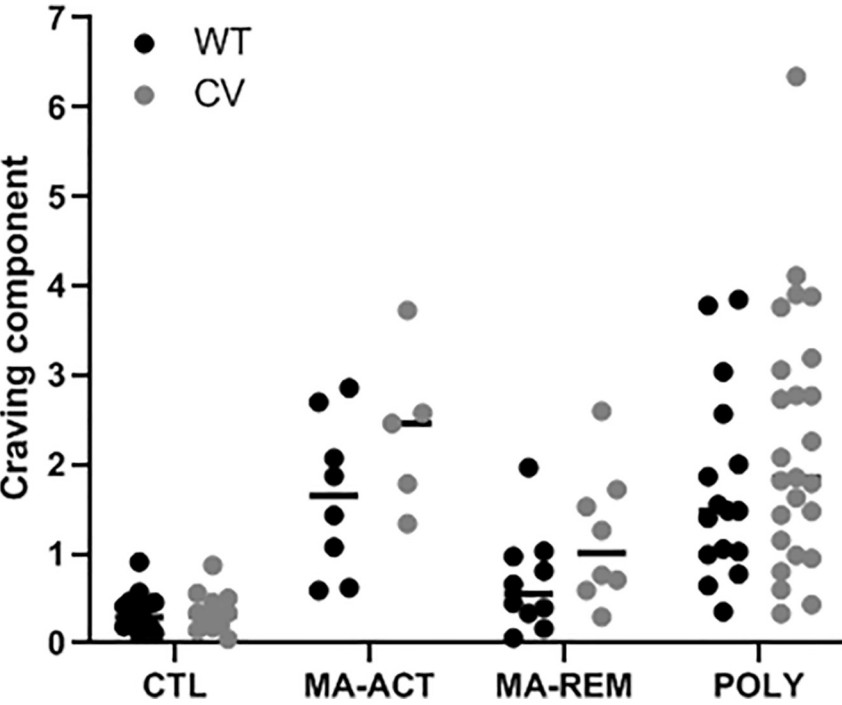

**Fig 2. Illustrates the craving response (vertical axis), as found by PCA and separated according to study group and *TAAR1* status (horizontal axis). Individual values are shown as black (WT = no V288V polymorphism) or gray (CV = homozygous or heterozygous for V288V polymorphism) circles, with median value indicated by horizontal dash.** For individuals with a history of methamphetamine (MA) dependence, the *TAAR1* positive response was ~1.68 times the negative response, indicating that participants with the *TAAR1* V288V SNP reported higher levels of craving (for alcohol, MA, and other addictive drugs) than participants without the *TAAR1* V288V SNP. Abbreviations: ACT, active; CV, common variant; POLY, polysubstance; REM, remission; WT, wild-type.

not known whether this effect occurs *in vivo*. While the V288V SNP is synonymous, such changes can affect the binding sites for splicing regulatory proteins and have significant effects on pre-mRNA splicing and translation efficiency [35, 36]. As *TAAR1* has a single exon, the increased protein production demonstrated in **Fig 1** is likely due to increased translation.

*TAAR1* genotype has substantial effects on responses to amphetamines. In preclinical studies, MA drinking lines of mice, bred for high or low voluntary MA intake, and TAAR1 knockout mice demonstrate a major impact of the *Taar1* gene on several MA-related responses (*e.g.*, MA consumption, MA-induced conditioned taste aversion and conditioned place preference, and MA-induced hypothermia [37, 38]). A mouse model of TAAR1 over-expression reported increased spontaneous dopaminergic neuron firing rate and extracellular DA but blunted extracellular dopamine and hyper-locomotion response to acutely administered amphetamine [39].

Although other simulants (*e.g.*, cocaine and methylphenidate) and alcohol reliably increase dopamine release in the striatum, amphetamines are unique substrates of the dopamine transporter (DAT) that can interact with intracellular TAAR1 [40]. Acute amphetamine, *via* agonism at the intracellular TAAR1 receptor, causes internalization of the DAT [41, 42] and a glutamate transporter (excitatory amino acid transporter 3, EAAT3) [43]. Chronic MA exposure causes long-term decreases in DAT in animals [44, 45] and humans [46, 47]. It is possible that individuals with a history of MA use who are homozygous or heterozygous with the V288V polymorphism (and presumptive TAAR1 overexpression) may have even lower DAT numbers than WT individuals, greater dopamine persistence in the synapse in ventral

tegmental area and nucleus accumbens ventral striatum and thus greater craving due to the influence of these regions on corticolimbic targets [48]).

Interestingly, the *TAAR1* gene does not occur in genetic databases referencing addiction or neuropsychiatric disorders associated with dopaminergic pathology. These include databases generated from studies on: 1) subjective responses to amphetamines [49], 2) self-reported alcohol consumption [50], 3) cocaine dependence [51], 4) schizophrenia [17, 52], and 5) Parkinson's disease [53]. A recent genome-wide association study (GWAS) that investigated alleles that correlated with shared risk for alcohol (n = 521 participants), heroin (n = 1026 participants), and MA (n = 1749 participants) did not identify *TAAR1* [54]. This report is important as one of the only GWAS investigation of MA addiction, but the number of individuals with MA use disorder is still rather small and may miss genes of small effect or those that are necessary but not sufficient to confer risk. The lack of association with other substances of abuse is not entirely surprising as MA is a substrate of the DAT and an agonist at the intracellular TAAR1 receptor, while alcohol, cocaine, cannabinoids, and opioids are not transported into the cell nor do they act at the receptor. We suggest that the CV exerts its influence in MA use disorder after the individual becomes addicted to MA and does not confer risk for developing addiction *per se*. Therefore, we do not find it surprising that the CV, which exerts its influence through over expression of TAAR1, is only associated with the disorder after chronic exposure to MA. The Hart et al. (2012) paper deserves separate mention, as that study investigated subjective response to amphetamine in healthy volunteers and did not identify the CV as explaining any of the variance in subjective experience. This accords with our hypothesis that the CV exerts its behavioral effects (*e.g.*, craving) only after addiction has developed in MA use disorder and may, therefore, not affect subjective response in those without a history of MA exposure.

This study has some limitations. Current nicotine and caffeine users were not excluded. Smoking, and to a lesser extent, caffeine use, is common in substance using populations and exclusion of these subjects would have resulted in recruitment of an atypical sample. Smoking was, however, distributed evenly between the V288V and WT subgroups and was thus unlikely to have affected the conclusions. In addition, the sample size for this investigation is small for genetic studies. Nevertheless, there was a statistically significant moderate effect (d = 0.58) of genotype on craving demonstrated in the MA using groups. As this effect size is associated with a power of about 0.5 for this number of subjects, it is not unlikely that the effect was detected. Confirmation of the genotype effect with larger samples would increase confidence in this finding. Also, as there were only two V288V homozygotes in the MA groups, if was not possible to evaluate a possible gene dose effect.

## Conclusions

This study is the first study to report on the effects of a polymorphism of the *TAAR1* gene on characteristics of humans with a MA use disorder, but it is not without limitations, including the use of a cross-sectional study design which did not allow for definitive conclusions on causality and small group sample sizes that may have limited our statistical power to some extent. More studies are necessary to address these limitations and to expand our understanding of the effects of *TAAR1* genotype on neuroadaptations that result from chronic MA. To the extent that craving is associated with dopamine receptor stimulation [55], individuals with the CV and a history of chronic use would experience greater dopamine persistence in the synapse and thus greater craving. Elucidation of the mechanism of this effect is needed, as the TAAR1 is increasingly being described as a promising therapeutic target for neuropsychiatric disorders

[13, 56, 57], and SNPs in *TAAR1* could provide a tool for individualizing treatments to improve early intervention strategies for the treatment of MA use disorder.

## Supporting information

**S1 Statistical Supplement. Calculation of effect size of *TAAR1* genotype and *post-hoc* power are summarized and discussed.**
(DOCX)

## Acknowledgments

We are grateful to the staffs of the VAPORHCS Substance Abuse Treatment Program, CODA Treatment Recovery, De Paul Treatment Centers, Central City Concern, Native American Rehabilitation Association, Outside In, and Volunteers of America Residential Treatment Centers, Portland, OR for their recruitment efforts and help with this study. We thank Sarah Anderson, Matthew Arbuckle, Alissa Bazinet, John Conley, and Bethany Winters for their contributions to data collection and study coordination. We also thank Dr. Clive Woffendin and the Oregon Health & Science University Clinical and Translational Research Center for sequencing the *TAAR1* gene and Elaine Huang for processing and cryopreserving the blood samples.

This work was supported in part by NIAAA R21AA020039 (WFH), Department of Veterans Affairs Clinical Sciences Research and Development Merit Review Program award number I0CX001558 (WFH), Department of Veterans Affairs Biomedical Laboratory Research and Development Merit Review Program award numbers 1 I01 BX002061 (JML) and 5 I01 BX002758 (AJ), DOJ 2010-DD-BX-0517 (WFH), NIDA P50 DA018165 (WFH, JML, MH, AJ), the Oregon Clinical and Translational Research Institute (OCTRI), grant number UL1TR002369 from the National Center for Advancing Translational Sciences (NCATS), a component of the National Institutes of Health (NIH), and NIH Roadmap for Medical Research. AJ is supported by the VA Research Career Scientist Program.

## Author Contributions

**Conceptualization:** Jennifer M. Loftis, Marilyn Huckans.

**Data curation:** Jennifer M. Loftis.

**Formal analysis:** Michael Lasarev, William F. Hoffman.

**Investigation:** Jennifer M. Loftis, Xiao Shi.

**Resources:** Aaron Janowsky.

**Supervision:** Jodi Lapidus, Aaron Janowsky, Marilyn Huckans.

**Writing – original draft:** Jennifer M. Loftis.

**Writing – review & editing:** Aaron Janowsky, William F. Hoffman, Marilyn Huckans.

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
