## [Decision Letter · Decision Letter 0]

16 Aug 2019

PONE-D-19-19007

Trace amine-associated receptor gene polymorphism increases drug craving

PLOS ONE

Dear Dr. Loftis,

Thank you for submitting your manuscript to PLOS ONE. After careful consideration, we feel that it has merit but does not fully meet PLOS ONE’s publication criteria as it currently stands. Therefore, we invite you to submit a revised version of the manuscript that addresses the points raised during the review process.

Please respond to the reviewers' comments particularly to methodological issues like a small sample size

We would appreciate receiving your revised manuscript by Sep 30 2019 11:59PM. To enhance the reproducibility of your results, we recommend that if applicable you deposit your laboratory protocols in protocols.io, where a protocol can be assigned its own identifier (DOI) such that it can be cited independently in the future. For instructions see: http://journals.plos.org/plosone/s/submission-guidelines#loc-laboratory-protocols

We look forward to receiving your revised manuscript.

Kind regards,

Aviv M. Weinstein

Academic Editor

PLOS ONE

Journal Requirements:

1. We note that you have indicated that data from this study are available upon request. PLOS only allows data to be available upon request if there are legal or ethical restrictions on sharing data publicly. For more information on unacceptable data access restrictions, please see http://journals.plos.org/plosone/s/data-availability#loc-unacceptable-data-access-restrictions.

Reviewers' comments:

Reviewer's Responses to Questions

**Comments to the Author**

1. Is the manuscript technically sound, and do the data support the conclusions?

Reviewer #1: Yes

Reviewer #2: Partly

2. Has the statistical analysis been performed appropriately and rigorously? 

Reviewer #1: Yes

Reviewer #2: Yes

3. Have the authors made all data underlying the findings in their manuscript fully available?

Reviewer #1: No

Reviewer #2: No

4. Is the manuscript presented in an intelligible fashion and written in standard English?

Reviewer #1: Yes

Reviewer #2: Yes

5. Review Comments to the Author

Reviewer #1: The paper "Trace amine-associated receptor gene polymorphism increases drug craving" by Loftis et al. investigated a common variant (CV) in the human TAAR1 gene, synonymous single nucleotide polymorphism (SNP) V288V, to determine the involvement of TAAR1 in methamphetamine addiction.

Participants with active meth dependence, in remission from meth dependence, with active poly-substance dependence, in remission from poly-substance dependence, and with no history of substance dependence completed neuropsychiatric symptom questionnaires and provided biological samples. Additionally, in vitro expression and function of CV and wild type TAAR1 receptors were also measured.

Results:

• Production of cAMP was higher in the CV- compared to WT-transfected cells in response to beta-PEA, but EC50 values did not differ. However, there was a significant increase in the maximal cAMP response to agonist stimulation in the cells expressing the CV, compared to the cells expressing the WT receptor.

• The V288V polymorphism had 40% increase in TAAR1 protein expression in cell culture, but message sequence and protein function were unchanged, suggesting an increase in translation efficiency.

• Principal components analysis resolved neuropsychiatric symptoms into four components, PC1 (depression, anxiety, memory, and fatigue), PC2 (pain), PC3 (drug and alcohol craving), and PC4 (sleep disturbances). Analyses of study group and TAAR1 genotype revealed a significant interaction for the “craving response” (PC3).

• The control group showed no difference in the “craving response” associated with TAAR1, while adjusted mean craving for the meth-dependence and meth-remission groups, among those with at least one copy of V288V, was 1.55 and 1.77 times, respectively, the adjusted mean craving for those without the TAAR1 SNP.

Based on their results the authors suggest that neuroadaptation to chronic MA use may be influenced by TAAR1 genotype and result in increased dopamine signalling and craving in individuals with the V288V genotype.

General comments:

This is an interesting research that focuses on putative genetic variants of methamphetamine addiction, which has a high impact on individual’s health and substantial burden for the society. The paper is in general well written. The methodology utilized and statistical analyses performed sounds adequate and the results provided may be of great interest for researchers in the field of the genetic determinants of drug addictions, in particular for psychostimulants, opening a door on the development of innovative pharmacological therapies having the TAAR1 receptor as a target for these pathological conditions. The reference list covers relevant and in-time research. The discussion of the results is adequate and well inserted in the contest of the current research in the field. It is appreciable that the authors recognize some of the major limits of that work (i.e., cross-sectional study, small sample).

Actually, I do not have substantial criticism on this interesting and well conducted work. However, the authors can find below a short list of suggestions/questions that, in my opinion, if addressed may strengthen the paper.

Some additional comments:

1. The title is someway misleading. In accordance with the data obtained, I suggest to change it in “Trace amine-associated receptor gene polymorphism increases drug craving in methamphetamine dependent individuals” or something similar, accounting for the fact that the significant association with craving has been found only in meth users after addiction has developed.

2. It would be better provide the reader of a short description of the neuropsychological tests used instead of redirecting them to the cited references.

3. Due to the strong association between psychostimulants addictions and psychotic symptoms, I am wondering why a neuropsychological test on psychotic symptoms (or something similar) has not been included in the study?

4. Some explanation on the exclusion of nicotine and caffeine as addictive drugs in the inclusion/exclusion criteria should be provided. In particular, nicotine is a potent addictive drug and several genetic variants were found to be linked to its addiction and relapse.

5. Some comment on the putative brain areas involved in the observed differences and putatively responsible for the higher craving in the CV group could be provided to the reader.

Reviewer #2: The present manuscript by Loftis and colleagues describes a series of translational experiments looking at the influence of a SNP in the TAAR1. Lab studies with transfected CHO cells demonstrated an effect of the polymorphism on maximal cAMP response to agonist stimulation. In a parallel study in humans, with five groups, the polymorphism was associated with an increase in a factor for drug craving in subjects who were current or previous methamphetamine users, by not in polydrug users. The was no influence of the SNP on other neuropsychiatric factors, such as mood or sleep.

Overall, the results of the study are interesting, and novel too - as the authors point out that this is the first study of the association between this SNP and methamphetamine craving in humans. The authors have made a good attempt to conduct a translational study, as they demonstrate modest effects of this synonymous polymorphism on agonist effects on cAMP levels.

However, the study has a number of limitations, which should be addressed. These are listed in the general order that they appear in the manuscript:

1/ The first sentence starts with a bang, with an attention-grabbing statement about the growing use of meth; but the reference included may not be the most scientifically valid one. Perhaps include an additional reference (e.g.https://www.ncbi.nlm.nih.gov/pubmed/23273775) to complement the existing one.

2/ Also in the introduction, the authors state that "Polymorphisms in several genes are associated with drug

dependence, including genes encoding opioid receptors..." The authors should probably specify which drugs are associated with these polymorphisms.

3/ In the Methods, were the controls recruited systematically from a different location than the drug users? The authors don't seem to mention SES, which may be a potential concern.

4/ How was history of prior medical illness determined? Was it by self-report - if so, please specify?

5/ Details about the urine drug screens should be provided, as they are not all the same.

6/ Importantly, who conducted the interviews to determine if subjects met criteria for DSM-IV diagnoses, and how were they qualified and/or trained? If multiple interviewers were used, do we know their inter-rater reliability?

7/ Please provide more details about the VAS scales used to measure craving. Are these standardized scales that have been validated?

8/ Where were procedures (including interviews and venipuncture) conducted?

9/ A bit more detail should be provided about the nonlinear regression used to analyze CHO data.

10/ In the Results, the authors refer to the variable of "race". I believe that "ethnicity" is now the more preferred term.

11/ Was there a gene dose response: in other words, did homozygotes for the SNP show greater responses?

12/ Probably the only major concern about this study is the relatively small sample size for the groups (including as few as 11 and 13 in two groups). This is obviously small for a genetic study...the authors are still able to eke out an effect for the PC3 factor on craving. The authors need to address this in more detail. It is briefly mentioned as a limitation, but it needs additional discussion - potentially including a power analysis of some form. I think that - on balance - the rigor used to separate the groups, combined with the detailed phenotype of the subjects, has led to a study of interest. But this last limitation is important, and should be addressed further.

6. PLOS authors have the option to publish the peer review history of their article (what does this mean?). If published, this will include your full peer review and any attached files.

Reviewer #1: No

Reviewer #2: No

---

## [Author Response · Author response to Decision Letter 0]

12 Sep 2019

August 28, 2019

Dear Dr. Weinstein,

The reviewers of our manuscript provided helpful comments, and we hope you agree that our responses have substantially improved our submission.

Reviewer #1: 

“This is an interesting research that focuses on putative genetic variants of methamphetamine addiction, which has a high impact on individual’s health and substantial burden for the society. The paper is in general well written. The methodology utilized and statistical analyses performed sounds adequate and the results provided may be of great interest for researchers in the field of the genetic determinants of drug addictions, in particular for psychostimulants, opening a door on the development of innovative pharmacological therapies having the TAAR1 receptor as a target for these pathological conditions. The reference list covers relevant and in-time research. The discussion of the results is adequate and well inserted in the contest of the current research in the field. It is appreciable that the authors recognize some of the major limits of that work (i.e., cross-sectional study, small sample). Actually, I do not have substantial criticism on this interesting and well conducted work. However, the authors can find below a short list of suggestions/questions that, in my opinion, if addressed may strengthen the paper.”

We thank the reviewer for these encouraging general comments and below have responded point-by-point to the specific comments.

1. “The title is someway misleading. In accordance with the data obtained, I suggest to change it in ‘Trace amine-associated receptor gene polymorphism increases drug craving in methamphetamine dependent individuals’ or something similar, accounting for the fact that the significant association with craving has been found only in meth users after addiction has developed.”

We like the suggestion of a more specific title and accepted the suggestion.

2. “It would be better provide the reader of a short description of the neuropsychological tests used instead of redirecting them to the cited references.”

Brief descriptions of the neuropsychological tests for those not face-valid, in addition to references, have been provided in the revised manuscript (please see Procedures).

3. “Due to the strong association between psychostimulants addictions and psychotic symptoms, I am wondering why a neuropsychological test on psychotic symptoms (or something similar) has not been included in the study?”

We used the MINI structured interview to identify psychotic symptoms. We also note that subjects with active psychosis were excluded. This has been clarified in the Materials and Methods section.

4. “Some explanation on the exclusion of nicotine and caffeine as addictive drugs in the inclusion/exclusion criteria should be provided. In particular, nicotine is a potent addictive drug and several genetic variants were found to be linked to its addiction and relapse.”

Nicotine and caffeine use are ubiquitous in substance-using populations. Excluding subjects who used these drugs would have eliminated all of the MA group subjects. We have added to the Discussion section to note that the frequency of smoking did not differ between V288V and wild type groups.

5. “Some comment on the putative brain areas involved in the observed differences and putatively responsible for the higher craving in the CV group could be provided to the reader.”

We hesitate to speculate excessively on specific brain regions, as we did not measure any in vivo imaging in this report. We note, however, in the discussion that TAAR1 has a prominent role on dopamine projections to the striatum and within the ventral tegmental area and that these regions influence the neurocircuitry of craving.

Reviewer #2: 

“Overall, the results of the study are interesting, and novel too - as the authors point out that this is the first study of the association between this SNP and methamphetamine craving in humans.”

We thank the reviewer for their positive comments and below have responded point-by-point to the specific comments.

1. “The first sentence starts with a bang, with an attention-grabbing statement about the growing use of meth; but the reference included may not be the most scientifically valid one. Perhaps include an additional reference (e.g.https://www.ncbi.nlm.nih.gov/pubmed/23273775) to complement the existing one.”

We agree that some peer-reviewed epidemiological references would be helpful and strengthen the paper. Accordingly, we have added two sentences and three references to the Introduction.

2. “Also in the introduction, the authors state that ‘Polymorphisms in several genes are associated with drug dependence, including genes encoding opioid receptors...’ The authors should probably specify which drugs are associated with these polymorphisms.”

A detailed discussion of addiction genetics is beyond the scope of this report. We have included, in addition to the specific examples with opioid addiction, a reference to a review of addiction genetics (i.e., Agrawal A, Edenberg HJ, Gelernter J. Meta-Analyses of Genome-Wide Association Data Hold New Promise for Addiction Genetics. J Stud Alcohol Drugs. 2016;77(5):676-80).

3. “In the Methods, were the controls recruited systematically from a different location than the drug users? The authors don't seem to mention SES, which may be a potential concern.”

Controls were recruited, primarily on-line or through newspaper listings, from the same Portland metro area as the individuals with MA dependence. We excluded subjects with greater than an associate’s degree to lessen differences in socioeconomic status. This information has been added to the revised manuscript. 

4. “How was history of prior medical illness determined? Was it by self-report - if so, please specify?”

Subjects’ medical histories were determined by self-report, now noted in the revised manuscript.

5. “Details about the urine drug screens should be provided, as they are not all the same.”

We thank the reviewer for raising this point. Urine drug screens were performed with Uscreen 6 panel Drug Test Cups (RapidDetectINC, Poteau, OK) (please see revised Methods section). The test screens for cocaine, marijuana, opioids, amphetamine, methamphetamine and benzodiazepines and was read for qualitative results on-site. No confirmation tests were performed.

6. “Importantly, who conducted the interviews to determine if subjects met criteria for DSM-IV diagnoses, and how were they qualified and/or trained? If multiple interviewers were used, do we know their inter-rater reliability?”

The following has been added to ‘Procedures’: Interviews were conducted by trained research associates under the direction of neuropsychologist Marilyn Huckans, PhD. Diagnoses and interpretation of responses to structured interviews were discussed at weekly meetings with Dr. Huckans, but no formal inter-rater reliability measures were calculated.

7. “Please provide more details about the VAS scales used to measure craving. Are these standardized scales that have been validated?”

More information has been added about the VAS scales used, including a reference which reports on the correlations between single-item VAS and a multiple-question Likert-type scale. 

8. “Where were procedures (including interviews and venipuncture) conducted?”

Interviews and venipuncture were conducted at the VA Portland Health Care System. This information has been added to ‘Procedures’.

9. “A bit more detail should be provided about the nonlinear regression used to analyze CHO data.”

A reference for the standard curve fitting software, GraphPad Prism (San Diego, CA), is provided in the revised manuscript. 

10. “In the Results, the authors refer to the variable of "race". I believe that "ethnicity" is now the more preferred term.” 

Thank you for this feedback. “Race” has been changed to “ethnicity”.

11. “Was there a gene dose response: in other words, did homozygotes for the SNP show greater responses?”

The reviewer raises a good question. There were only two CV homozygotes in the MA group, and this was insufficient to determine a gene dose effect. 

12. “Probably the only major concern about this study is the relatively small sample size for the groups (including as few as 11 and 13 in two groups). This is obviously small for a genetic study...the authors are still able to eke out an effect for the PC3 factor on craving. The authors need to address this in more detail. It is briefly mentioned as a limitation, but it needs additional discussion - potentially including a power analysis of some form. I think that - on balance - the rigor used to separate the groups, combined with the detailed phenotype of the subjects, has led to a study of interest. But this last limitation is important, and should be addressed further.”

The effect size (Cohen’s d) for genotype in the multiple regression model was moderate (0.58). As this corresponded to a post hoc power of approximately 0.5, we do not think it was unlikely that we detected this effect, even though our sample size was small. We added the calculation of effect size to the Results section and commented on sample size in this context in the Discussion section. In addition, we added post hoc power calculations as Supplementary Information.

Thank you again for the careful review of our manuscript.

Sincerely,

Jennifer M. Loftis, Ph.D.

---

## [Decision Letter · Decision Letter 1]

30 Sep 2019

Trace amine-associated receptor gene polymorphism increases drug craving in individuals with methamphetamine dependence

PONE-D-19-19007R1

Dear Dr. Loftis,

We are pleased to inform you that your manuscript has been judged scientifically suitable for publication and will be formally accepted for publication once it complies with all outstanding technical requirements.

With kind regards,

Aviv M. Weinstein

Academic Editor

PLOS ONE

Additional Editor Comments (optional):

Reviewers' comments:

Reviewer's Responses to Questions

**Comments to the Author**

1. If the authors have adequately addressed your comments raised in a previous round of review and you feel that this manuscript is now acceptable for publication, you may indicate that here to bypass the “Comments to the Author” section, enter your conflict of interest statement in the “Confidential to Editor” section, and submit your "Accept" recommendation.

Reviewer #1: All comments have been addressed

Reviewer #2: All comments have been addressed

2. Is the manuscript technically sound, and do the data support the conclusions?

Reviewer #1: Yes

Reviewer #2: Yes

3. Has the statistical analysis been performed appropriately and rigorously? 

Reviewer #1: Yes

Reviewer #2: Yes

4. Have the authors made all data underlying the findings in their manuscript fully available?

Reviewer #1: (No Response)

Reviewer #2: No

5. Is the manuscript presented in an intelligible fashion and written in standard English?

Reviewer #1: Yes

Reviewer #2: Yes

6. Review Comments to the Author

Reviewer #1: (No Response)

Reviewer #2: (No Response)

7. PLOS authors have the option to publish the peer review history of their article (what does this mean?). If published, this will include your full peer review and any attached files.

Reviewer #1: No

Reviewer #2: No

---

## [Editor Report · Acceptance letter]

2 Oct 2019

PONE-D-19-19007R1 

Trace amine-associated receptor gene polymorphism increases drug craving in individuals with methamphetamine dependence 

Dear Dr. Loftis:

I am pleased to inform you that your manuscript has been deemed suitable for publication in PLOS ONE. Congratulations! Your manuscript is now with our production department. 

With kind regards,

on behalf of

Dr. Aviv M. Weinstein 

Academic Editor

PLOS ONE